# OpenReview forum: "No, Of Course I Can! Refusal Mechanisms Can Be Exploited Using Harmless Data"
_ICLR.cc/2025/Workshop/BuildingTrust — BuildingTrust_

### Official Review · Reviewer_BVJS · 2025-02-16
**Organize existing jailbreaks, suggest effective defenses, and propose an attack (NOICE) that can bypass the defense**

**Rating:** 6
**Confidence:** 4

**Review:**

# Summary

- This paper first summarizes recent attacks (in particular CMF and ISA), noting that many of them exploit the model's tendency to make a harmful response when the answer begins with a helpful prefix. As a result, many such attacks target the adjustment of the initial few tokens.
- As a defense against attacks with such characteristics, the paper proposes generating the first k-tokens using an aligned model (AMD) or forcibly adding a refusal at the beginning (FRD). This significantly mitigates the attack success rate.
- Furthermore, it suggests an attack that bypasses these defenses by training a model such that it initially appears to refuse appropriately, but then generates harmful responses afterward. This method, called "No, Of course I Can Execute" (NOICE), achieves a high ASR by circumventing their proposed defenses (AMD and FRD).

# Strength

- The threat model considers realistic scenarios by incorporating constraints presented by OpenAI and Google, such as training data volume, cost, and policy.
- Experiments conducted on multiple model families, model sizes, and even production fine-tuning APIs, demonstrating the attack's success in diverse settings.
- The proposed defenses (FRD, AMD) effectively counter attacks based on existing attacks (YOC, ISA).
- The proposed attack (NOICE) successfully bypasses FRD, AMD, and even llama-guard (LG).

# Weakness

- Table 6 is missing: It seems there is only a caption.
- Ability to bypass other existing defenses: While the paper presents three defenses, the proposed attack seems to inherently bypass FRD and AMD as these defenses assume that only the initial few tokens need to be considered for the attack's success. Although bypassing LG (which is used in practice) is impressive, it would be beneficial to further organize the discussion by (i) examining which other existing defenses the attack can bypass and which it cannot, and (ii) proposing defenses that could potentially counter NOICE if existing ones are ineffective.

---

### Official Review · Reviewer_uvaY · 2025-03-01
**Comments**

**Rating:** 5
**Confidence:** 4

**Review:**

Strength:
1. The jailbreak problem of open-source models.
2. Further analysis of the impact of both prefix rejection and acceptance on subsequent responses.

Weakness:
1. The paper’s threat model claims a data poisoning assumption for closed-source models, strictly limiting both the quantity and stealthiness of the poisoned data. However, only Table 4 presents experiments on closed-source models, while other most experiments are conducted on open-source models. This may be inconsistent with the paper’s motivation and claims. In particular, for open-source models, strict limitations on the amount and stealthiness of poisoned data in data poisoning may not be practical.
2. A baseline is lacked, as fine-tuning with harmless data itself might affect ASR.

---

### Official Review · Reviewer_f9Ax · 2025-03-04
**Simple and insightful approach for fine-tuning models to break safety training**

**Rating:** 8
**Confidence:** 4

**Review:**

### Summary

This work shows how LLMs (open source AND proprietary models) are vulnerable to fine-tuning attacks even when restricted to (relatively) harmless data by leveraging the fact that many alignment techniques focus on some sort of refusal in the first few output tokens, and circumventing this by allowing the model to refuse, and then have the model output a harmful response. This attack can be done with harmless data, which at the time of publication (now presumably patched) was able to bypass API content filters on harmful fine-tuning training data. This attack is effective against both open sourced models as well as proprietary models offering fine-tuning APIs, such as ChatGPT.

### Strengths

- The authors propose a simple and insightful approach for fine-tuning models to break safety training that is effective against a variety of modern LLMs
- Good overview and implementation of SFT attack baselines
- Strong and comprehensive experimental results, especially on closed sourced models (ChatGPT, Claude)

### Weaknesses

- As with many works published on LLM jailbreaks/safety, attacks/defenses are likely to become obsolete with some future updates to the model training pipelines/APIs (i.e. a “rat race”).  However given the nature of the venue, I believe the insights and results presented here are relevant and useful for the community, and does not warrant scrutiny for this potential shortcoming.

### Questions

- One set of results that I would be interested in seeing is how the component of the refusal direction (Arditi et al. 2024) varies under such an attack. It would be very interesting if a refusal direction is still present in the model’s latent space prior to the affirmative response.

---

### Decision · Program_Chairs · 2025-03-04

Accept